# The Impact of Everolimus and Radiation Therapy on Pulmonary Fibrosis

**DOI:** 10.3390/medicina56070348

**Published:** 2020-07-13

**Authors:** Mehmet Fuat Eren, Ayfer Ay Eren, Mutlay Sayan, Birsen Yücel, Şahende Elagöz, Yıldıray Özgüven, Irina Vergalasova, Ahmet Altun, Saadettin Kılıçkap, Vasudev Malik Daliparty, Nuran Beşe

**Affiliations:** 1Radiation Oncology Clinic, M.H. Marmara University Pendik Education and Research Hospital, 34899 İstanbul, Turkey; 2Radiation Oncology Clinic, Kartal Dr. Lütfi Kırdar Education and Research Hospital, 34890 İstanbul, Turkey; drayferay@gmail.com; 3Department of Radiation Oncology, Rutgers Cancer Institute of New Jersey, New Brunswick, NJ 08901, USA; ms2641@cinj.rutgers.edu (M.S.); irinav@cinj.rutgers.edu (I.V.); 4Department of Radiation Oncology, Cumhuriyet University School of Medicine, 58140 Sivas, Turkey; yucelbirsen@yahoo.com; 5Department of Pathology, Cumhuriyet University School of Medicine, 58140 Sivas, Turkey; selagoz65@gmail.com; 6Department of Medical Physics, Trakya University School of Medicine, 22030 Edirne, Turkey; yildq@hotmail.com; 7Department of Pharmacology, Cumhuriyet University School of Medicine, 58140 Sivas, Turkey; md.ahmetaltun@gmail.com; 8Department of Preventive Oncology, Hacettepe University School of Medicine, 06230 Ankara, Turkey; skilickap@yahoo.com; 9Department of Internal Medicine, Raritan Bay Medical Center, Perth Amboy, NJ 08861, USA; vasudaliparty@gmail.com; 10Department of Radiation Oncology, Medicine Faculty, Acibadem University, 34457 Istanbul, Turkey; nuranbese@superonline.com

**Keywords:** radiation therapy, rat, everolimus, pneumonitis, pulmonary fibrosis

## Abstract

*Background and objectives:* Everolimus (EVE) is a mammalian target of the rapamycin (mTOR) inhibitor that is widely used in cancer patients. Pulmonary toxicity, usually manifesting as interstitial pneumonitis, is a serious adverse effect of this drug. Radiation therapy, which is often administered in conjunction with chemotherapy for synergistic effects, also causes pulmonary fibrosis. In view of pulmonary damage development in these two forms of cancer treatment, we have examined the effect of EVE administration individually, in combination with radiation given in varying sequences, and its relation to the extent of pulmonary damage. *Materials and Methods:* We performed an experimental study in albino rats, which were randomized into five groups: (1) control group, (2) EVE alone, (3) EVE 22 h after radiation, (4) EVE 2 h after irradiation, and (5) only radiation. Sixteen weeks after thoracic irradiation, rat lung tissue samples were examined under light microscopy, and the extent of pulmonary damage was estimated. After this, we calculated median fibrosis scores in each group. *Results:* The highest fibrosis score was noted in Group 4. Among the five groups, the control group had a significantly lower median fibrosis score compared to the others. When the median fibrosis score of the group that received concurrent EVE with radiation therapy (RT) (Group 4) was compared with that of the control group, the difference was statistically significant (*p* = 0.0022). However, no significant differences were achieved among the study groups that received EVE only or RT only, whether concurrently or sequentially (*p* > 0.05). *Conclusion:* EVE is an effective treatment option for the management of several malignancies and is often combined with other therapies, such as radiation, for a more efficient response. However, an increased risk of pulmonary fibrosis should also be anticipated when these two modalities are combined, as they both can cause pulmonary damage, especially when administered concurrently.

## 1. Introduction

Several novel targeted molecular therapies are being used with increased efficacy in the management of malignancies. Among them, everolimus (EVE) is an agent that acts via inhibition of the mammalian target of rapamycin (mTOR), a protein kinase which regulates cell growth and protein synthesis in response to various biological and mechanical stimuli [1,2]. This process results in disruption of metabolic homeostasis, leading to a halt in the translation of genes that regulate cancer cell proliferation. EVE is utilized in the management of breast cancer, neuroendocrine tumors and renal cell carcinoma, among others [3]. EVE is known to be relatively safe; however, pulmonary fibrosis has been noted in several clinical trials as a rare yet serious adverse effect of this drug [4,5].

Another mode of treatment used in the management of malignancies is radiation, which can be used as a curative, adjuvant or palliative form of therapy. Ionizing radiation causes damage to both rapidly proliferating malignant cells and also to normal tissue within the radiation field. This damage leads to inflammation, causing stimulation of fibroblasts differentiation into myofibroblasts, subsequently leading to tissue remodeling and fibrosis [6]. The lung is one of the most radiosensitive organs, and it is frequently irradiated as part of treatment programs for cancers of the lung, esophagus, breast, and lymphatic system. Radiation-induced pulmonary toxicity is a common and critical problem that limits the doses that can be delivered. The incidence of symptomatic radiation-induced pulmonary toxicity may be as high as 30%, and a higher risk is expected for patients treated with combined chemoradiotherapy, a higher total dose, a larger fraction size, and a larger volume of irradiated lung [7,8,9].

In view of pulmonary toxicity as an adverse effect in both EVE administration and radiotherapy, in this study we have aimed to understand the effect of EVE, when used alone or in combination with concurrent or sequential radiation therapy, on lung tissue.

## 2. Materials and Methods

### 2.1. Animals

Thirty female Wistar albino rats [10,11] initially weighing 200–210 g were produced, bred, and housed in the Experimental Animal Breeding and Research Laboratory at Cumhuriyet University. Six animals were housed per cage and maintained under identical conditions with food and water provided ad libitum. All experiments were carried out in compliance with the regulations of our institution and the 3R (reduction, replacement, refinement) ethical guidelines and ethical approval was obtained from the local Experimental Animal Research Ethical Committee (No:01042013/371).

Rats were randomized into five experimental groups, each housed in separate cages. Group 1 (Control, *n* = 6) included rats which did not receive any treatment. Group 2 (EVE only, *n* = 6) consisted of rats, which received only EVE. Group 3 (Sequential RT-EVE, *n* = 6) included rats, which received thoracic irradiation followed by administration of EVE after a gap of 22 h. The rationale for this dose was based on the reasoning that the half-life of EVE is 21 h and 22 h in plasma and tumor tissue, respectively [12]. Group 4 (Concurrent EVE, *n* = 6) consisted of rats that received EVE followed by thoracic irradiation within 2 h of drug administration. Group 5 (radiation therapy (RT) only, *n* = 6) received only thoracic irradiation (Table 1).

### 2.2. Administration

EVE (1.5 mg/kg, Afinitor^®^; Novartis, Basel, Switzerland) was administered via the intraperitoneal (ip) route following dilution with 2 cc of 0.9% NaCl solution. Rats were anesthetized with an intramuscular injection of 2% xylazine hydrochloride (3 mg/kg, Rompun^®^; Bayer Kimya San. Ltd. Sti., Istanbul, Turkey) and ketamine hydrochloride (90 mg/kg, Ketalar^®^; EWL Eczacibasi Warner Lambert Ilac¸ Sanayi ve Ticaret A.S., Istanbul, Turkey) prior to simulation and irradiation. The animals were securely held in a foam holder in a supine position, and plastic bandages were used to immobilize the thoracic region during irradiation. A 0.5 cm elastic-gel bolus was used to provide contour regularity (Figure 1A–C). All six groups, excluding Group 1 were irradiated to the whole thoracic region with a 6 MV linear accelerator (Varian Clinac^®^ DHX). A single dose of 12 Gy was administered to both lungs with a 4 × 4 cm anterior single field at 2 cm depth using the Source Axis Distance technique [13].

### 2.3. Tissue Preparations

Rats were anesthetized and sacrificed 16 weeks after RT, which was shown to be a sufficient period for the development of radiation-induced lung fibrosis in rats [14]. Lung tissue samples were fixed by tracheal instillation of 10% neutral-buffered formalin, embedded in paraffin and 4-um sections were cut. To assess fibrosis in each group, sections were stained with Masson’s trichrome, and examination of the slides was carried out under light microscopy. We tested fibrosis based on alveolar septal thickening with superimposed collagen. The area adjacent to the large bronchi and vessels was estimated at 20× magnification. As a quantitative end point, the extent of pulmonary fibrosis was graded on a scale of 0 (normal lung or minimal fibrous thickening as shown in Figure 2A) to 4 (total fibrous obliteration of the field as shown in Figure 2E) as described in Table 2. The pathologist was not aware of the treatment groups at the time of the histological examination of the specimens. After examining the entire sections for each rat, the average value was taken as the fibrosis score, and mean values of the groups were calculated.

### 2.4. Statistics

Continuous variables were reported as medians and interquartile ranges (IQR) since non-normality was observed by Kolmogorov–Smirnov and Shapiro–Wilks tests. Kruskal–Wallis analyzes of variance (ANOVA) and the Tamhane post-hoc test were carried out to test for differences in means among treatment groups [15], and a significance level of 0.05 was considered significant. Graphpad prism Software version 8.3.0 and IBM SPSS Statistics (Version 23, IBM, Armonk, NY, USA) were used for statistical analysis.

## 3. Results

At the conclusion of the study (16 weeks after thoracic irradiation was given), all rats included in the study were alive. In these 30 rats, histopathological analysis was performed on lung tissue samples, and lung fibrosis in each sample was assessed. The median fibrosis scores (IQR) for each group using the number of surviving rats and histopathological analysis was calculated and given in Table 3. The highest median fibrosis score was noted in the concurrent EVE group (Group 4), which received EVE followed by thoracic irradiation within 2 h of drug administration (Figure 2). The Kruskal–Wallis test was used to gauge for statistically significant differences in fibrosis scores between distinct groups. Among the five groups, the control group had a significantly lower median fibrosis score compared to the rest of the groups. When the median fibrosis score of the group that received concurrent EVE with RT (Group 4) was compared with that of the control group (Group 1), the difference was statistically significant (*p* = 0.0022), as shown in Figure 3. It was observed that EVE and RT caused an increase in fibrosis scores. However, no significant differences were achieved among the study groups that received EVE only or RT only, whether concurrently or sequentially administered (*p* > 0.05).

## 4. Discussion

The use of targeted molecular therapies in oncology has been on the rise, owing to the high efficacy of these drugs. EVE is an mTOR inhibitor that has emerged as an effective drug in the management of malignancies, such as breast cancer, renal cell carcinoma, subependymal giant cell astrocytoma, and neuroendocrine tumors. However, as with all mTOR inhibitors, EVE carries a troublesome adverse effect profile. The frequently reported adverse effects seen with EVE use are fatigue, rash, stomatitis, nausea, and diarrhea. A more serious adverse effect that has been noted is pulmonary toxicity, which can manifest in the form of interstitial lung damage [5,16]. The exact underlying mechanism of this toxicity has not yet been clearly understood.

Autoimmune mechanisms and delayed hypersensitivity reactions have been implicated as possible causes for pulmonary fibrosis. A study conducted by Fielhaber et al., hypothesized that mTOR inhibitors would suppress apoptosis genes, such as STAT-1, which could lead to lipopolysaccharide-induced pulmonary damage [17]. A recently conducted in vitro identification study suggested that the pulmonary fibrosis induced by EVE may be partly caused by the transition of airway cells from epithelial to mesenchymal tissue and has been implicated in complicated biological networks in this transformation [18]. The pulmonary toxicity of EVE has also been clinically demonstrated. A phase-three clinical trial conducted by Motzer et al. in 416 patients with renal cell carcinoma showed an incidence of non-infectious pneumonitis in 14% of the study group [19]. The BOLERO-2 trial conclusively proved that the addition of EVE to exemestane provided increased efficacy in hormone receptor-positive breast cancer treatment and showed that pneumonitis occurred more commonly with EVE addition compared to the placebo arm [20].

Radiation therapy is an essential treatment option used in the management of malignancies. It is often administered with chemotherapy, owing to its synergistic effect. Radiation therapy can be administered for a curative, adjunctive and a palliative purpose based on the clinical scenario [21]. Ionizing radiation can cause DNA damage, which can lead to cell death by destroying the neoplastic cells. Radiotherapy also causes adverse effects of an acute, long-term, or cumulative nature. Radiation-induced pulmonary fibrosis is one such long-term, serious adverse effect. The underlying mechanism of this pulmonary damage is very complex and involves several signaling pathways at the molecular level. Ionizing radiation triggers changes at the cellular level, involving endothelial cells, pneumocytes and immune cells, which lead to remodeling and fibrotic changes in the lung parenchyma [22].

Radiation dose, the volume of lung parenchyma irradiated, and the concomitant administration of chemotherapeutic agents are significant factors that determine the extent of lung tissue damage. A study conducted by Manegold et al. concluded that pulmonary endothelial cells and lung tissue were most sensitive to therapy when a combination of EVE and radiotherapy were administered [23]. The underlying basis of this synergistic effect was attributed to a combined antivascular and antiangiogenic mechanism. Another experimental study showed that a combination of B-cell lymphoma 2 (Bcl-2) and mTOR inhibition led to increased radiosensitization in a lung cancer models [24]. In 2012, a meta-analysis of published series, which is the largest report of the incidence and the relative risk to develop pulmonary fibrosis in patients treated with EVE or temsirolimus for several types of cancer, showed that incidence of high-grades pulmonary toxicity was 2.4% [25]. Furthermore, in a similar experimental study performed by Bese et al. [26], the concurrent and sequential use of tamoxifen with pulmonary radiation therapy were evaluated in Wistar albino rats. The highest pulmonary fibrosis scores were obtained in the concurrent group. Again, in our study, we showed that concurrent use of EVE and radiation therapy increases the pulmonary fibrosis score compared to the other study groups. However, statistical analysis proved the difference to be insignificant except when compared with the control group. It is presumed that the current number of subjects (six rats per group) is insufficient to derive a statistically significant result; this small sample number seems to be the most significant limitation of our study.

This synergistic effect of EVE and radiation therapy provides a more effective treatment option for successful management of malignancies, but it also carries the possibility of an increased risk of lung fibrosis. We attempted to examine the combination of these two treatments, both concurrently and sequentially, to obtain a clear picture of the differing extent of lung parenchyma damage in each scenario.

## 5. Conclusions

At this point, while concurrent EVE administration increases pulmonary fibrosis with radiation therapy, safety results of randomized trials should be used to evaluate the ideal time of EVE and radiation therapy administration.

## Figures and Tables

**Figure 1 medicina-56-00348-f001:**
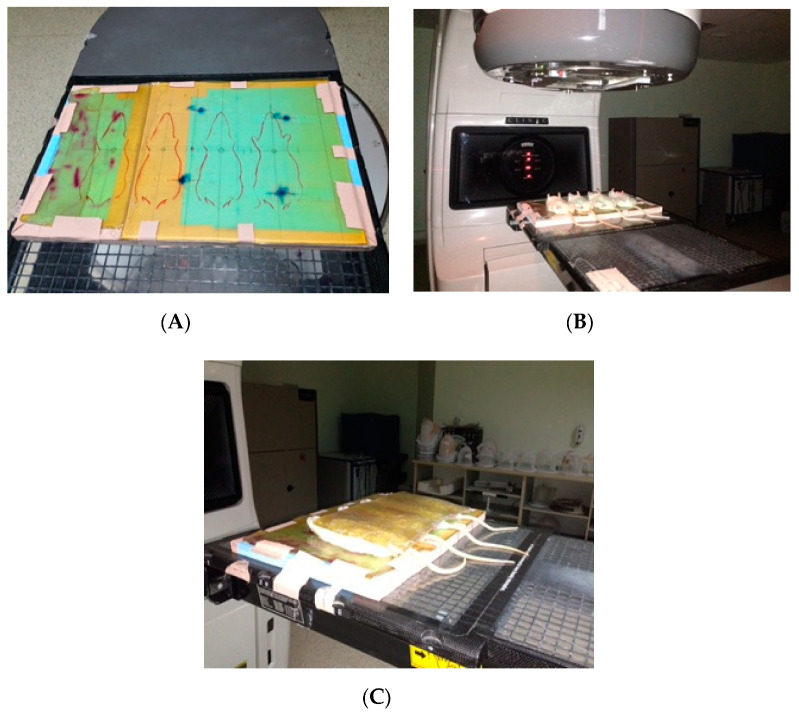
The animals were held securely on a foam holder (**A**) in a supine position, and plastic bandages were used to immobilize the thoracic region during irradiation (**B**). An elastic-gel bolus (0.5 cm) was used to provide contour regularity (**C**).

**Figure 2 medicina-56-00348-f002:**
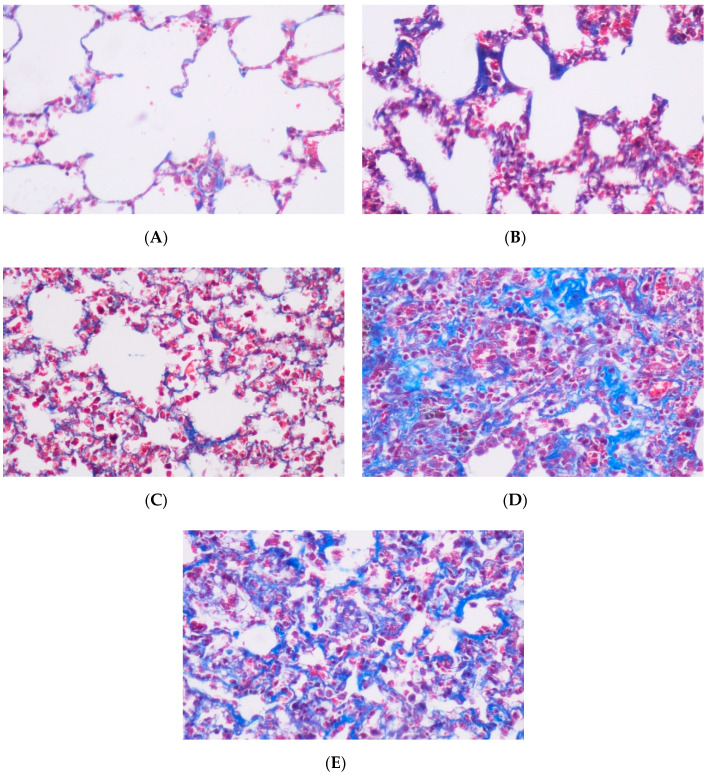
Grade 0, normal lung (Masson’s trichrome × 100) (**A**); Grade 1, isolated alveolar septa with gentle fibrotic changes (Masson trichrome × 200) (**B**); Grade 2, fibrotic changes of alveolar septa with knot-like formation (Masson trichrome × 200) (**C**); Grade 3, contiguous fibrotic walls of alveolar septa (Masson trichrome × 100) (**D**); and Grade 4, single fibrotic mass (Masson trichrome × 100) (**E**).

**Figure 3 medicina-56-00348-f003:**
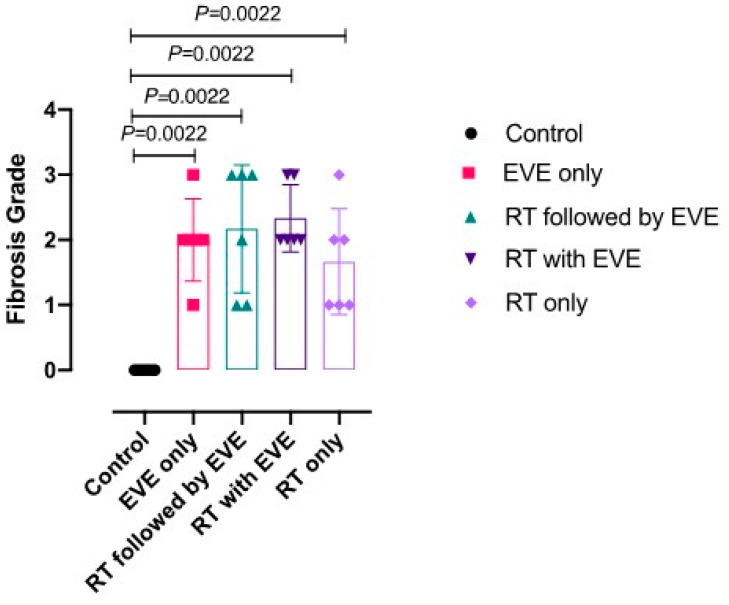
Median fibrosis score of the treatment groups compared to the control group. EVE: Everolimus. RT: Radiotherapy.

**Table 1 medicina-56-00348-t001:** The abbreviations used for the study groups.

Group (G)	
G1	Control group
G2	Everolimus only group
G3	RT + sequential everolimus group
G4	RT + concurrent everolimus group
G5	RT only group

G: group; RT: radiotherapy.

**Table 2 medicina-56-00348-t002:** Criteria for grading lung fibrosis.

Grade	Histological Features
0	Normal lung or minimal fibrous thickening of alveolar or bronchial walls.
1	Moderate thickening of the wall without obvious damage to lung architecture.
2	Increased fibrosis with definitive damage to lung structure and formation of fibrous bands or small fibrosis masses.
3	Severe distortion of the structure and large fibrous areas; ‘‘honeycomb lung’’ is placed in this category.
4	Total fibrous obliteration of the field.

**Table 3 medicina-56-00348-t003:** The distribution of animals according to their study groups and median fibrosis scores for each group ^a^.

Study Groups	Median Fibrosis Score (IQR)
Group 1, Control (*n* = 6)	0,00 (0.0, 0.00)
Group 2, EVE only (*n* = 6)	2.00 (1.0, 2.0) ^b^
Group 3, Sequential RT-EVE (*n* = 6)	2.00 (1.75, 3.00) ^b^
Group 4, Concurrent EVE (*n* = 6)	2.50 (2.00, 3.00) ^b^ < 0.05
Group 5, RT only (*n* = 6)	2.00 (1.75, 2.25) ^b^

^a^ ANOVA test and Tamhane post-hoc test were carried out to test for differences in means among study groups, and a significance level of 0.05 was considered significant. ^b^
*p* < 0.05 when compared to control group. EVE: Everolimus. RT–EVE: Radiotherapy–Everolimus.

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
