# Peer review of "The Impact of Everolimus and Radiation Therapy on Pulmonary Fibrosis"

_medicina, 2020, doi:10.3390/medicina56070348_

Round 1

Reviewer 1 Report

They reported about the relationship between everolimus and radiation. Their content is interesting, but I have some confirmation.

  1. In Line 90, Line 101, and Line 121, They mention the figure, but I can not find all figures. I can not judge the report for the absence of the figure. If you forget to submit, please add the figure.

  1. In Line 122, they described that there was no significant difference other than in the control group, and also described the same statement in the Conclusion section. But there is no mention of them in Abstract. Conclusion in the text should be correspondence with conclusion in Abstract.

  1. It is a predictable result that a significant difference occurs between Group 1 and other groups. But, it is interesting that the analysis in the Concurrent group and the Sequential group was found no difference between them. It may be a more interesting report by adding the interpretation of the author.

  1. I think that the enrolled number in this study is small. It may be necessary to describe limitations.

Author Response

Cover Letter

We thank the editor(s) and all reviewers for evaluating our study. All the comments have significantly helped us improve the manuscript. We did an extensive revision of the original submission based on the critiques raised by each reviewer. We hope that the current version of the manuscript can now be found suitable for publication in the "Medicina" Journal. Below are the point by point responses to the critiques.

We also attach a clean revised version of the manuscript and a separate word document that shows the revisions as a track changes feature of word.

Reviewer #1:

Response:

Thank you for the comments and contributions. We have now made extensive revision to our manuscript to improve the overall quality of the paper, which we hope can be accepted in its revised form.

Comments:

  1. In Line 90, Line 101, and Line 121, They mention the figure, but I can not find all figures. I can not judge the report for the absence of the figure. If you forget to submit, please add the figure.

C1: I uploaded the figures to the system seperately so I added the figures 1-2-3 at the end of the manuscript.

Figure 1: The animals were held securely on a foam holder (A) in a supine position and plastic bandages were used to immobilize the thoracic region during irradiation (B). An elastic-gel bolus (0.5 cm) was used to provide contour regularity (C).

(A)      B)      C)

Figure 2: Grade 0, normal lung (Masson’s trichrome ×100) (A), Grade 1, isolated alveolar septa with gentle fibrotic changes (Masson trichrome ×200) (B), Grade 2, fibrotic changes of alveolar septa with knot-like formation (Masson trichrome ×200) (C), Grade 3, contiguous fibrotic walls of alveolar septa (Masson trichrome ×100) (D), and Grade 4, single fibrotic mass (Masson trichrome ×100) (E).

  1. A) B) C) D) E)

Figure 3: Median fibrosis score of the treatment groups compared to the control group.

  1. In Line 122, they described that there was no significant difference other than in the control group, and also described the same statement in the Conclusion section. But there is no mention of them in Abstract. Conclusion in the text should be correspondence with conclusion in Abstract.

C2: We added the sentence in abstract “result” section (line 39). ‘However, no significant differences were achieved among the study groups that received EVE only, RT only, whether concurrently or sequentially (p>0,05).’ Also I changed the conclusion in the text (line 171). ‘For now while concurrent EVE administiration increases pulmonary fibrosis with RT, safety results of randomised trials should be awaited for the ideal timing of EVE and RT administiration.’

  1. It is a predictable result that a significant difference occurs between Group 1 and other groups. But, it is interesting that the analysis in the Concurrent group and the Sequential group was found no difference between them. It may be a more interesting report by adding the interpretation of the author.

C3: We added a sentence about the analysis of concurrent and sequential groups in “discussion” (line 164-165). ‘ Again in the current study, we showed that concurrent use of EVE and RT increases the pulmonary fibrosis score compared to other groups. However, statistical analysis proved the difference to be insignificant except for the control group. It is presumed that current number of subjects (6 rats per group) is insufficient to deive a statistically significant result.’

  1. I think that the enrolled number in this study is small. It may be necessary to describe limitations.

C4: We described limitations in “discussion” section (line 165). ‘However, statistical analysis proved the difference to be insignificant except for the control group. It is presumed that current number of subjects (6 rats per group) is insufficient to derive a statistically significant result and this seems to be the most significant limitation of our study.’

Reviewer 2 Report

Interesting topic and question of study. However, more work should be performed before concluding that EVE does not affect fibrosis scores. 

  1. The English grammar and sentence structure needs to be modified. Please consider consulting with a company to help polish up the structure to improve the language. For example, "histological analysis was done" would sound more formal as "was performed." Another example, line 49 - "which are being used" - this sentence is wordy and can be restructured to be more concise. In the conclusion, "did not significantly increased" should be "increase." Also in the references, it looks like numbers are listed twice. 
  2. In table 2, the p values comparing these differences should be included. It would also be great to include figures representing the histopathological analysis (images) showing the different fibrosis scores and examples per experimental group.
  3. The amount of data presented is sparse and not robust. Are there any other phenotypes observed in these mice? What other phenotypes do other works publish when analyzing fibrosis? 
  4. How does the administering of EVE and RT coincide with what is given to human beings. Why was a 16 week mark chosen? It would be great to include rationale and a description for how this mimics treatment and fibrosis development in human beings.
  5. The discussion states that fibrosis may be a result of auto immune mechanisms  - have the authors considered performing studies in mice with altered immune systems? Perhaps the incidence of fibrosis is not high in healthy mice, thus resulting in no significant differences observed before the experimental groups analyzed. 

I think this is an interesting study and the authors are on the right path, but more work needs to be done to solidify the conclusions behind made. 

Author Response

Cover Letter

We thank the editor(s) and all reviewers for evaluating our study. All the comments have significantly helped us improve the manuscript. We did an extensive revision of the original submission based on the critiques raised by each reviewer. We hope that the current version of the manuscript can now be found suitable for publication in the "Medicina" Journal. Below are the point by point responses to the critiques.

We also attach a clean revised version of the manuscript and a separate word document that shows the revisions as a track changes feature of word.

Reviewer #2:

Response:

Thank you for the comments and contributions. We have now made extensive revision to our manuscript to improve the overall quality of the paper, which we hope can be accepted in its revised form.

  1. The English grammar and sentence structure needs to be modified. Please consider consulting with a company to help polish up the structure to improve the language. For example, "histological analysis was done" would sound more formal as "was performed." Another example, line 49 - "which are being used" - this sentence is wordy and can be restructured to be more concise. In the conclusion, "did not significantly increased" should be "increase." Also in the references, it looks like numbers are listed twice.

C1: The English grammar and sentence structure were modified. It was shown in the main text.

  1. In table 2, the p values comparing these differences should be included. It would also be great to include figures representing the histopathological analysis (images) showing the different fibrosis scores and examples per experimental group.

C2: The p values were included in table 2. Also, histopathological images were added at the end of the manuscript (figure 2).

Table 3. The distribution of animals according to their study groups and median fibrosis scores for each group.a

Study Groups

Median Fibrosis Score (IQR)

Group 1, Control (n = 6)                 

             0,00 (0.0, 0.00)

Group 2, EVE only (n = 6)             

             2.00  (1.0, 2.0)b

Group 3, Sequential RT– EVE (n = 6)

 2.00  (1.75, 3.00)b

Group 4, Concurrent EVE (n = 6)        

             2.50  (2.00, 3.00)b    < 0.05

Group 5, RT only (n = 6)               

 2.00  (1.75, 2.25)b

a ANOVA test and Tamhane post hoc test were carried out to test for differences in means among study groups and a significance level of 0.05 was considered significant.

b  p < 0.05 when compared to control group.

Figure 2: Grade 0, normal lung (Masson’s trichrome ×100) (A), Grade 1, isolated alveolar septa with gentle fibrotic changes (Masson trichrome ×200) (B), Grade 2, fibrotic changes of alveolar septa with knot-like formation (Masson trichrome ×200) (C), Grade 3, contiguous fibrotic walls of alveolar septa (Masson trichrome ×100) (D), and Grade 4, single fibrotic mass (Masson trichrome ×100) (E).

  1. A) B) C) D) E)

  1. The amount of data presented is sparse and not robust. Are there any other phenotypes observed in these mice? What other phenotypes do other works publish when analyzing fibrosis? 

C3: We described why we chose wistar albino rats with references (line 70). Moreover, most of the studies in radiation oncology uses this phenotype of rat.

References:

  1. Yavas G, Gultekin M, Yildiz O, et al. Assessment of concomitant versus sequential trastuzumab on radiation-induced cardiovascular toxicity. Hum Exp Toxicol. 2017;36(11):1121-1130. doi:10.1177/0960327116680276
  2. Altinok AY, Yildirim S, Altug T, et al. Aromatase inhibitors decrease radiation-induced lung fibrosis: Results of an experimental study. Breast. 2016;28:174-177. doi:10.1016/j.breast.2016.04.003
  3. Bese NS, Umay C, Serdengecti S, et al. The impact of trastuzumab on radiation-induced pulmonary fibrosis: results of an experimental study. Med Oncol. 2010;27(4):1415-1419. doi:10.1007/s12032-009-9395-5

  1. How does the administering of EVE and RT coincide with what is given to human beings. Why was a 16 week mark chosen? It would be great to include rationale and a description for how this mimics treatment and fibrosis development in human beings.

C4: We changed the sentence in the line 94, and we explained ‘why was a 16 week mark chosen?’with a reference. “Rats were anesthetized and sacrificed with cervical dislocation 16 weeks after RT which was shown to be a sufficient period for the development of radiation-induced lung fibrosis in rats [12]”. Also, we added a description and a meta-analysis in discussion for fibrosis development in human beings (line 165). ‘In 2012, a meta-analysis of published series, which is the largest report of the incidence and the relative risk to develop pulmonary fibrosis in patients treated with everolimus or temsirolimus for several types of cancer, showed that incidence of high-grades pulmonary toxicity was 2.4%.’

  1. The discussion states that fibrosis may be a result of auto immune mechanisms  - have the authors considered performing studies in mice with altered immune systems? Perhaps the incidence of fibrosis is not high in healthy mice, thus resulting in no significant differences observed before the experimental groups analyzed. 

C5: Pulmonary fibrosis was observed in all the animals who had RT and also It was observed that EVE and RT increased fibrosis scores. Pulmonary fibrosis was not observed in control group (healthy group) as you stated in your comment. Moreover, in radiation oncology, most of the experimental studies have been done in healthy rats.

Round 2

Reviewer 1 Report

The authors have sincerely revised the paper according to my comments.

Author Response

Cover Letter

We thank the editor(s) and all reviewers for evaluating our study. All the comments have significantly helped us improve the manuscript. We did an extensive revision of the original submission based on the critiques raised by each reviewer. We hope that the current version of the manuscript can now be found suitable for publication in the "Medicina" Journal. Below are the point by point responses to the critiques.

We also attach a clean revised version of the manuscript and a separate word document that shows the revisions as a track changes feature of word.

Reviewer #1:

Thank you for the comments and contributions. We have now made extensive revision to our manuscript to improve the overall quality of the paper, which we hope can be accepted in its revised form. Moreover, the manuscript was edited for grammar, language and style by American Manuscript Editors.

  1. INTRODUCTION

We added descriptive sentences at the end of the line 63. These sentences are about ‘why we choose the lung for the experimental studies?’

The lung is one of the most radiosensitive organs, and it is frequently irradiated as part of treatment programmes for cancers of the lung, oesophagus, breast and lymphatic system. Radiation-induced pulmonary toxicity is a common and critical problem that limits the doses that can be delivered. The incidence of symptomatic radiation-induced pulmonary toxicity may be as high as 30% and a higher risk is expected for patients treated with combined chemoradiotherapy, a higher total dose, a larger fraction size, and a larger volume of irradiated lung”

  1. MATERIALS-METHODS

Animals

We changed the sentences line 73-74-75, and we added new sentences.

Six animals were housed per cage and maintained under identical conditions with food and water provided ad libitum. All experiments were carried out in compliance with the regulations of our institution and the 3R (reduction, replacement, refinement) ethical guidelines and ethical approval was obtained from the local Experimental Animal Research Ethical Committee (No:01042013/371).”

Administration

We changed the sentences line 95-96-97, and we added new sentences.

All six groups, excluding group 1 were irradiated to the whole thoracic region with 6 MV linear accelerator (Varian ClinacⓇ DHX). A single dose of 12-Gy was administered to both lungs with a 4×4 cm anterior single field at 2 cm depth using Source Axis Distance technique”

  1. RESULTS

We added a new sentence at the end of the line 127.

“The median fibrosis scores (IQR) for each group using the number of surviving rats and histopathological analysis was calculated and given in Table 3.”

  1. REFERENCES

We added two new references ‘8 and 9’ which cited in introduction section.

“8. Vujaskovic Z, Marks LB, Anscher MS. The physical parameters and molecular events associated with radiation-induced lung toxicity. Semin Radiat Oncol. 2000 Oct;10(4):296-307. doi: 10.1053/srao.2000.9424.

  1. Bese NS, Munzuroglu F, Uslu B, Arbak S, Yesiladali G, Sut N, Altug T, Ober A. Vitamin E protects against the development of radiation-induced pulmonary fibrosis in rats. Clin Oncol (R Coll Radiol). 2007 May;19(4):260-4. doi: 10.1016/j.clon.2006.12.007. “

Reviewer 2 Report

I still am finding issues with the English writing. Can you please clarify that the entire manuscript was edited for grammar? I am also seeing different font sizes throughout the text.

Author Response

Cover Letter

We thank the editor(s) and all reviewers for evaluating our study. All the comments have significantly helped us improve the manuscript. We did an extensive revision of the original submission based on the critiques raised by each reviewer. We hope that the current version of the manuscript can now be found suitable for publication in the "Medicina" Journal. Below are the point by point responses to the critiques.

We also attach a clean revised version of the manuscript and a separate word document that shows the revisions as a track changes feature of word.

Reviewer #2:

Thank you for the comments and contributions. We have now made extensive revision to our manuscript to improve the overall quality of the paper, which we hope can be accepted in its revised form. Moreover, the manuscript was edited for grammar, language and style by American Manuscript Editors.

  1. INTRODUCTION

We added descriptive sentences at the end of the line 63. These sentences are about ‘why we choose the lung for the experimental studies?’

The lung is one of the most radiosensitive organs, and it is frequently irradiated as part of treatment programmes for cancers of the lung, oesophagus, breast and lymphatic system. Radiation-induced pulmonary toxicity is a common and critical problem that limits the doses that can be delivered. The incidence of symptomatic radiation-induced pulmonary toxicity may be as high as 30% and a higher risk is expected for patients treated with combined chemoradiotherapy, a higher total dose, a larger fraction size, and a larger volume of irradiated lung”

  1. MATERIALS-METHODS

Animals

We changed the sentences line 73-74-75, and we added new sentences.

Six animals were housed per cage and maintained under identical conditions with food and water provided ad libitum. All experiments were carried out in compliance with the regulations of our institution and the 3R (reduction, replacement, refinement) ethical guidelines and ethical approval was obtained from the local Experimental Animal Research Ethical Committee (No:01042013/371).”

Administration

We changed the sentences line 95-96-97, and we added new sentences.

All six groups, excluding group 1 were irradiated to the whole thoracic region with 6 MV linear accelerator (Varian ClinacⓇ DHX). A single dose of 12-Gy was administered to both lungs with a 4×4 cm anterior single field at 2 cm depth using Source Axis Distance technique”

  1. RESULTS

We added a new sentence at the end of the line 127.

“The median fibrosis scores (IQR) for each group using the number of surviving rats and histopathological analysis was calculated and given in Table 3.”

  1. REFERENCES

We added two new references ‘8 and 9’ which cited in introduction section.

“8. Vujaskovic Z, Marks LB, Anscher MS. The physical parameters and molecular events associated with radiation-induced lung toxicity. Semin Radiat Oncol. 2000 Oct;10(4):296-307. doi: 10.1053/srao.2000.9424.

9. Bese NS, Munzuroglu F, Uslu B, Arbak S, Yesiladali G, Sut N, Altug T, Ober A. Vitamin E protects against the development of radiation-induced pulmonary fibrosis in rats. Clin Oncol (R Coll Radiol). 2007 May;19(4):260-4. doi: 10.1016/j.clon.2006.12.007. “

Round 3

Reviewer 2 Report

No additional comments for the authors